# Efficient Diffusion Posterior Sampling for dose reduced CT reconstruction

## Abstract

The clinical efficacy of Computed Tomography (CT) is well-established, yet concerns regarding its radiation exposure persist. To mitigate this risk, a reduction in X-ray photon count or projection views is typically pursued, albeit at the expense of image quality. In this study, we introduce an innovative diffusion posterior sampling approach for CT image reconstruction at reduced radiation doses. This method initiates with a predictive step, leveraging data enhancement on the posterior approximation derived from a pre-trained diffusion model and the measurement data. Subsequently, a forward sampling phase ensues, which maps the output to a noisy timestep, followed by a diffusion estimation process. Additionally, we propose an acceleration strategy that employs superior initialization to significantly curtail the sampling steps required. Our experimental findings indicate that this method not only enhances the quality of reconstructed images by an average of 3.5 db but also accelerates the process to over ten times faster than existing diffusion-based techniques. These outcomes underscore the method's potential in clinical settings.

## 1 Introduction

Computed tomography (CT) imaging technology has undergone significant advancements in the past decades, leading to substantial improvements in diagnostic performance. The CT reconstruction problem can be written as the following ill-posed inverse problem:

$$y = Ax + n \tag{1}$$

where $y, x$ denote the measured projection and the unknown CT image to be reconstructed; $A$ is the forward projection operator and $n$ is the measurement noise which is often modelled by i.i.d. random variables. However, the increasing use of medical CT has raised concerns about potential radiation risks, including genetic damage and cancer. Consequently, there is a pressing need to reduce radiation doses. The two primary strategies for dose reduction are lowering the X-ray tube current and decreasing the number of scanning views (Slovis, 2002). Nonetheless, these approaches inevitably introduce noise and artifacts in the reconstructed images, which can significantly degrade image quality and complicate accurate clinical diagnosis.

In existing literature, sinogram filtering methods, such as structural adaptive filtering (Balda et al., 2012), bilateral filtering (Manduca et al., 2009), and penalized weighted least-squares (Wang et al., 2006), have been proposed to enhance LDCT image quality by applying filters to sinogram data. However, due to low signal-to-noise ratios, these methods often fail to produce high-quality images. Iterative reconstruction methods address this by optimizing an objective function that incorporates prior knowledge from both the sinogram and image domains. Regularization techniques, such as total variation (TV) regularization (Sidky & Pan, 2008), wavelet-based sparsity priors (Jia et al., 2011), nonlocal total variation (Jia et al., 2010), and low-rank patch priors (Cai et al., 2014), are commonly used in these approaches. These sparse regularization models can be efficiently solved using first-order methods like ADMM (Boyd et al., 2011) or the Split Bregman method (Goldstein & Osher, 2009), enhancing CT image quality by leveraging statistical properties and prior knowledge.

In recent years, deep learning (DL) methods have attracted significant attention due to their impressive capabilities in reconstructing LDCT images. DL-based image postprocessing techniques utilize deep neural networks (DNNs) as denoisers to eliminate artifacts in images reconstructed by conventional filtered back projection (FBP) methods. Typical networks include FBPConvNet (Jin et al., 2017),

REDCNN(Chen et al., 2017), EDCNN (Liang et al., 2020) and so on. As the artifacts in the reconstructed images often cannot be modeled as independent random noise, the performance gain brought by postprocessing method is limited. The optimization unrolling scheme (Adler & Öktem, 2018; Ding et al., 2020; Chen et al., 2018) and plug-and-play methods (Ye et al., 2018) are indicated more effective approaches. Unlike traditional post-processing denoising networks, these methodologies derive their foundation from iterative reconstruction processes. By unrolling iterative algorithms or incorporating pre-trained denoising sub-networks, these methods facilitate the design of advanced deep reconstruction networks. However, these methods heavily depend on paired low and normal dose images for supervised learning, which is both challenging and costly in clinical settings. Acquiring such paired data increases workload, expenses, and raises concerns about patient safety and privacy. Additionally, these approaches often produce over-smoothed or hallucinated images, which can obscure critical details or introduce errors, potentially leading to misdiagnoses by radiologists.

Recently, the score-based diffusion model (Ho et al., 2020; Song et al., 2021c) has garnered significant attention due to its innovative approach to generative modeling, which effectively reconstructs data by progressively denoising purely gaussian noise. The properties of the diffusion model, such as its robustness to mode collapse and its capacity for generating high-fidelity outputs, indicate its potential in the field of image generation. Diffusion models have also shown remarkable progress in solving inverse problems, including super-resolution (Saharia et al., 2022), image inpainting (Song et al., 2021c), Magnetic Resonance Image (MRI) reconstruction (Song et al., 2021b), and CT reconstruction (Song et al., 2021b; Chung et al., 2022). Many of these works preserve the original training process but modify the inference procedure to enable sampling from a conditional distribution. This kind of approach utilizes the pre-trained score function as a generative prior for the data distribution, thus avoiding the need for paired data. Such flexibility allows for application across various tasks while maintaining superior reconstruction quality. However, achieving high-quality reconstruction generally requires a large number of diffusion posterior sampling steps, typically 1000 or 2000 steps as noted in (Chung et al., 2022; Song et al., 2021b). This requirement leads to significant computational costs due to repeated forward-backward operator evaluations and inference steps, which limits the efficiency in solving inverse problems. In imaging inverse problems, it is crucial to reduce noise and artifacts in the final image while preserving sharpness. The inherent randomness of diffusion models can introduce unpredictable elements into the reconstructed image if insufficient projection constraints are applied.

To enhance both the efficiency and quality of CT image reconstruction using diffusion model, we introduce an Efficient Diffusion Posterior Sampling (EffiDPSRecon) scheme. This method integrates diffusion sampling as generative priors within the iterative reconstruction process. Unlike previous algorithms that rely on a single-step gradient descent, resulting in an inaccurate approximation of the posterior log-likelihood, our method utilizes the conjugate gradient algorithm, initialized with denoised data, to improve data fidelity and accelerate convergence. Additionally, we introduce a forward resampling step, mapping the denoised data back into the noisy data space before evaluating a diffusion step. To further enhance efficiency, we initialize the process by incorporating prior information from Filtered Backprojection (FBP) images, which reduces the required number of sampling steps. Experiments on dose-reduced CT reconstruction, including both low-dose noisy CT (LDCT) and undersampled (Sparse-view CT) data, demonstrate that our approach reduces the number of diffusion steps from 1000 to just 50, while maintaining high image fidelity, with an average PSNR improvement of 3.5 dB. These results highlight the effectiveness of the proposed EffiDPSRecon method for clinical applications.

The remainder of this paper follows this structure: Section 2 provides a concise overview of prior literature relevant to the subject. Section 3 outlines the specifics of the proposed methodology. Section 4 showcases the experimental evaluation and comparison to other methods. Ultimately, our conclusions are presented in Section 5.

## 2 BACKGROUND

**Diffusion models**    The Denoising Diffusion Probabilistic Model (DDPM) (Ho et al., 2020), commonly referred to as one of the most classic diffusion models, comprises both forward and reverse processes. The forward process defines a transformation path from a clear image $x_0$ to completely

random gaussian noise which is governed by the following Markov chain with $N + 1$ states:

$$q\left(\boldsymbol{x}_{1:N} \mid \boldsymbol{x}_0\right) = \prod_{t=1}^{N} q\left(\boldsymbol{x}_t \mid \boldsymbol{x}_{t-1}\right) \tag{2}$$

Here, $q(\boldsymbol{x}_t|\boldsymbol{x}_{t-1})$ is a Gaussian distribution defined by:

$$q\left(\boldsymbol{x}_t \mid \boldsymbol{x}_{t-1}\right) = \mathcal{N}\left(\boldsymbol{x}_t \mid \sqrt{1-\beta_t}\boldsymbol{x}_{t-1}, \beta_t \boldsymbol{I}\right), \tag{3}$$

where $\beta_t \in [0, 1]$ is an increasing schedule which controls the noise level. Using the properties of the Gaussian distribution, one can directly sample $\boldsymbol{x}_t$ with given $\boldsymbol{x}_0$ as follows:

$$q\left(\boldsymbol{x}_t \mid \boldsymbol{x}_0\right) = \mathcal{N}\left(\boldsymbol{x}_t \mid \sqrt{\bar{\alpha}_t}\boldsymbol{x}_0, (1-\bar{\alpha}_t)\boldsymbol{I}\right), \tag{4}$$

for $\alpha_t = 1 - \beta_t$ and $\bar{\alpha}_t = \prod_{i=1}^{t} \alpha_i$. As $t$ gradually increases, $x_N$ finally becomes noise following standard Gaussian distribution. The training loss of DDPM is designed for noise prediction in the reverse sampling process:

$$\mathcal{L} = \mathbb{E}_{\boldsymbol{x}_0}\mathbb{E}_{\boldsymbol{\epsilon}\sim\mathcal{N}(0,\boldsymbol{I}),t}\left\|\boldsymbol{\epsilon} - \boldsymbol{\epsilon}_\theta\left(\sqrt{\bar{\alpha}_t}\boldsymbol{x}_0 + \sqrt{1-\bar{\alpha}_t}\boldsymbol{\epsilon}, t\right)\right\|_2^2. \tag{5}$$

Starting from random gaussian noise $\boldsymbol{x}_T$, the reverse sampling of DDPM can be written as:

$$\boldsymbol{x}_{t-1} = \frac{1}{\sqrt{\alpha_t}}\left(\boldsymbol{x}_t - \frac{\beta_t}{\sqrt{1-\bar{\alpha}_t}}\boldsymbol{\epsilon}_\theta\left(\boldsymbol{x}_t, t\right)\right) + \sigma_t\boldsymbol{z}. \tag{6}$$

Song et al. (2021c) further proposed a unified framework which transforms the DDPM from discrete-time formulation to continuous-time counterpart through stochastic differential equation (SDE). The corresponding SDE for the forward process of DDPM can be formulated as:

$$\mathrm{d}\boldsymbol{x} = -\frac{1}{2}\beta(t)\boldsymbol{x}\mathrm{d}t + \sqrt{\beta(t)}\mathrm{d}\boldsymbol{w}, \tag{7}$$

where $\beta(t)$ represents the noise schedule of the forward process, corresponding to the continuous form of $\beta_t$, and $\boldsymbol{w}$ is the standard Wiener process. Then the reverse SDE for sampling is:

$$d\boldsymbol{x} = \left[-\frac{\beta(t)}{2}\boldsymbol{x} - \beta(t)\nabla_{\boldsymbol{x}}\log p_t\left(\boldsymbol{x}\right)\right]dt + \sqrt{\beta(t)}d\overline{\boldsymbol{w}} \tag{8}$$

where $\overline{\boldsymbol{w}}$ is the Wiener process for the reverse SDE and the term $\nabla_{\boldsymbol{x}}\log p_t(\boldsymbol{x})$ is the score function, which is approximated by a neural network $\boldsymbol{s}_\theta(\boldsymbol{x}_t, t)$ in practice. The connection between score function $\boldsymbol{s}_\theta$ and noise prediction $\boldsymbol{\epsilon}_\theta$ in DDPM can be formulated approximately as $\boldsymbol{s}_\theta(\boldsymbol{x}_t, t) \approx -\frac{\boldsymbol{\epsilon}_\theta(\boldsymbol{x}_t, t)}{\sqrt{1-\bar{\alpha}_t}}$.

**Diffusion Denoising Implicit Model (DDIM)**   In order to sample with diffusion models more efficiently, Song et al. proposed DDIM (Song et al., 2021a), where the diffusion process can be extended from Markovian to non-Markovian and (6) can be rewritten as:

$$\boldsymbol{x}_{t-1} = \sqrt{\bar{\alpha}_{t-1}}\hat{\boldsymbol{x}}_0\left(\boldsymbol{x}_t\right) + \sqrt{1-\bar{\alpha}_{t-1}-\sigma_{\eta_t}^2}\cdot\boldsymbol{\epsilon}_\theta\left(\boldsymbol{x}_t, t\right) + \sigma_{\eta_t}\boldsymbol{\epsilon}_t, \tag{9}$$

where $\boldsymbol{\epsilon}_t$ is standard Gaussian noise and $\hat{\boldsymbol{x}}_0\left(\boldsymbol{x}_t\right)$ denotes the predicted $\boldsymbol{x}_0$ from $\boldsymbol{x}_t$ with Tweedie's formula:

$$\hat{\boldsymbol{x}}_0\left(\boldsymbol{x}_t\right) := E\left[\boldsymbol{x}_0 \mid \boldsymbol{x}_t\right] = \frac{1}{\sqrt{\bar{\alpha}_t}}\left(\boldsymbol{x}_t + (1-\bar{\alpha}_t)\nabla_{\boldsymbol{x}_t}\log p_t\left(\boldsymbol{x}_t\right)\right) \approx \frac{\boldsymbol{x}_t - \sqrt{1-\bar{\alpha}_t}\boldsymbol{\epsilon}_\theta\left(\boldsymbol{x}_t, t\right)}{\sqrt{\bar{\alpha}_t}}, \tag{10}$$

and the magnitude $\sigma_{\eta_t}$ of noise $\boldsymbol{\epsilon}_t$ controls how stochastic the diffusion process is. $\sigma_{\eta_t} = 0$ yields fully deterministic sampling while $\sigma_{\eta_t} = \sqrt{(1-\bar{\alpha}_{t-1})/(1-\bar{\alpha}_t)}\sqrt{1-\bar{\alpha}_t/\bar{\alpha}_{t-1}}$ yields the original sampling pattern of DDPM.

**Solving linear inverse problems with diffusion models.** Given measurements $\boldsymbol{y} \in \mathbb{R}^m$ and a forward measurement operator $\boldsymbol{A}$, diffusion models effectively address inverse problems by substituting the score function in Equation (8) with the conditional score function $\nabla_{\boldsymbol{x}_t} \log p(\boldsymbol{x}_t \mid \boldsymbol{y})$. By applying Bayes' rule, this conditional score can be expressed as:

$$\nabla_{\boldsymbol{x}_t} \log p_t(\boldsymbol{x}_t \mid \boldsymbol{y}) = \nabla_{\boldsymbol{x}_t} \log p_t(\boldsymbol{x}_t) + \nabla_{\boldsymbol{x}_t} \log p_t(\boldsymbol{y} \mid \boldsymbol{x}_t). \tag{11}$$

This decomposition supports the formulation of a reverse SDE as follows:

$$d\boldsymbol{x} = \left[ -\frac{\beta(t)}{2} \boldsymbol{x} - \beta(t) \left( \nabla_{\boldsymbol{x}_t} \log p_t(\boldsymbol{x}_t) + \nabla_{\boldsymbol{x}_t} \log p_t(\boldsymbol{y} \mid \boldsymbol{x}_t) \right) \right] dt + \sqrt{\beta(t)} d\overline{\boldsymbol{w}}. \tag{12}$$

With a pre-trained score diffusion model $\boldsymbol{s}_\theta (\boldsymbol{x}_t, t) \approx \nabla_{\boldsymbol{x}_t} \log p (\boldsymbol{x}_t)$, posterior sampling is achieved by merely modifying the sampling process. However, challenges arise primarily due to the lack of an analytical expression for the likelihood term $\nabla_{\boldsymbol{x}_t} \log p(\boldsymbol{y} \mid \boldsymbol{x}_t)$. To address this, researchers have explored two primary strategies: the first involves applying alternating projections onto the measurement subspace to avoid direct use of the likelihood such as Manifold Constraint Gradient (MCG) (Chung et al., 2022), and the second entails approximating the likelihood under reasonable assumptions (Chung et al., 2023). For example, Chung et al. (2023) developed a technique known as diffusion posterior sampling (DPS) with the following update steps:

$$\begin{aligned} \boldsymbol{x}'_{t-1} &= \sqrt{\bar{\alpha}_{t-1}} \hat{\boldsymbol{x}}_0 (\boldsymbol{x}_t) + \sqrt{1 - \bar{\alpha}_{t-1} - \sigma_{\eta_t}^2} \cdot \boldsymbol{\epsilon}_\theta (\boldsymbol{x}_t, t) + \sigma_{\eta_t} \boldsymbol{\epsilon}_t, \\ \boldsymbol{x}_{t-1} &= \boldsymbol{x}'_{t-1} - \zeta \nabla_{\boldsymbol{x}_t} \| \boldsymbol{y} - \boldsymbol{A}(\hat{\boldsymbol{x}}_0(\boldsymbol{x}_t)) \|_2^2, \end{aligned} \tag{13}$$

where $\zeta \in \mathbb{R}$ is a tunable step-size parameter. Despite these advancements, it is crucial to note that these methods sometimes struggle to perform effective posterior sampling and may exhibit slow convergence rates, which limits their practical application in real-world scenarios.

## 3 METHODS

In this section, we introduce our proposed method, namely Efficient Diffusion Posterior Sampling (EffiDPSRecon), which aim to improve the sampling speed and reconstruction quality of existing diffusion-based posterior sampling methods for inverse problems. The key idea of EffiDPSRecon is to incorporate measurement information at multiple stages of the diffusion sampling process to enhance data consistency and accelerate the sampling process with a FBP sampled initialization. In the following, we first present our method in steps and then draw connections and difference to existing DPS based methods.

### 3.1 EFFICIENT DIFFUSION POSTERIOR SAMPLING RECONSTRUCTION

Based on a pre-trained diffusion model on CT images, we aim to develop an efficient sampling method from observed projection measurements $\boldsymbol{y}$. The overall reconstruction procedure is illustrated in Fig. 1. In the following, we explain the procedure from $\boldsymbol{x}_t$ to $\boldsymbol{x}_{t-1}$ in detail.

**Posterior Mean Estimation** Considering the DDPM-based forward process $p_t(\boldsymbol{x}_t \mid \boldsymbol{x}_0) = \mathcal{N}(\boldsymbol{x}_t; \sqrt{\bar{\alpha}_t} \boldsymbol{x}_0, (1 - \bar{\alpha}_t) \boldsymbol{I})$, the posterior mean conditioned on $\boldsymbol{x}_t$ and $\boldsymbol{y}$ is given by:

$$\hat{\boldsymbol{x}}_0(\boldsymbol{x}_t, \boldsymbol{y}) := E[\boldsymbol{x}_0 \mid \boldsymbol{x}_t, \boldsymbol{y}] = \frac{1}{\sqrt{\bar{\alpha}_t}} (\boldsymbol{x}_t + (1 - \bar{\alpha}_t) \nabla_{\boldsymbol{x}_t} \log p_t(\boldsymbol{x}_t \mid \boldsymbol{y})), \tag{14}$$

which follows from the properties of Gaussian distributions and the application of Tweedie's formula.

Using the Bayes' rule from Equation (11), and substituting the pre-trained score approximation $\nabla_{\boldsymbol{x}_t} \log p_t(\boldsymbol{x}_t) \approx -\frac{\boldsymbol{\epsilon}_{\boldsymbol{\theta}}(\boldsymbol{x}_t, t)}{\sqrt{1 - \bar{\alpha}_t}}$, we obtain:

$$E[\boldsymbol{x}_0 \mid \boldsymbol{x}_t, \boldsymbol{y}] = \frac{1}{\sqrt{\bar{\alpha}_t}} (\boldsymbol{x}_t + (1 - \bar{\alpha}_t) \nabla_{\boldsymbol{x}_t} \log p_t(\boldsymbol{y} \mid \boldsymbol{x}_t) - \sqrt{1 - \bar{\alpha}_t} \, \boldsymbol{\epsilon}_{\boldsymbol{\theta}}(\boldsymbol{x}_t, t)). \tag{15}$$

Assuming Gaussian measurement noise, where $\boldsymbol{y} \sim \mathcal{N}(\boldsymbol{A}\boldsymbol{x}_0, \sigma^2 \boldsymbol{I})$, the gradient $\nabla_{\boldsymbol{x}_t} \log p_t(\boldsymbol{y} \mid \boldsymbol{x}_t)$ can be approximated by:

$$\nabla_{\boldsymbol{x}_t} \log p_t(\boldsymbol{y} \mid \boldsymbol{x}_t) \approx -\frac{1}{\sigma^2} \nabla_{\boldsymbol{x}_t} \| \boldsymbol{y} - \boldsymbol{A} \, \hat{\boldsymbol{x}}_0(\boldsymbol{x}_t) \|_2^2, \tag{16}$$

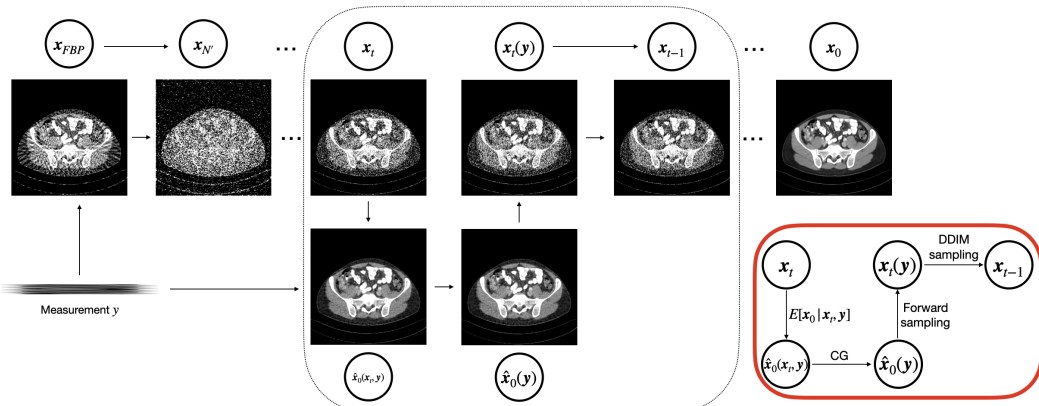

Figure 1: Overall diagram of EffiDPSRecon. The steps within the red box correspond to a detailed breakdown of one iteration from $\boldsymbol{x}_t$ to $\boldsymbol{x}_{t-1}$ in the overall procedure. The left section illustrates the transition between different time steps, starting from $x_{N'}$ to $x_0$.

where $\hat{\boldsymbol{x}}_0(\boldsymbol{x}_t)$ is defined from Tweedie's formula (10). Substituting this approximation back into the expression for the posterior mean estimate, we have:

$$\hat{\boldsymbol{x}}_0(\boldsymbol{x}_t, \boldsymbol{y}) \approx \hat{\boldsymbol{x}}_0(\boldsymbol{x}_t) - \rho_t \, \nabla_{\boldsymbol{x}_t} \left\| \boldsymbol{y} - \boldsymbol{A} \, \hat{\boldsymbol{x}}_0(\boldsymbol{x}_t) \right\|_2^2, \tag{17}$$

where $\rho_t$ is a step size parameter.

**Data Enhancement Projection** To further enhance data consistency, we project the updated posterior mean $\hat{\boldsymbol{x}}_0(\boldsymbol{x}_t, \boldsymbol{y})$ onto the data manifold defined by the normal equation:

$$\mathcal{M}_0 = \{\boldsymbol{x} \mid \boldsymbol{A}^T \boldsymbol{A} \boldsymbol{x} = \boldsymbol{A}^T \boldsymbol{y}\} \tag{18}$$

i.e.

$$\hat{\boldsymbol{x}}_0(\boldsymbol{y}) := \mathcal{P}_{\mathcal{M}_0}[\hat{\boldsymbol{x}}_0(\boldsymbol{x}_t, y)], \tag{19}$$

where $\mathcal{P}_{\mathcal{M}_0}$ denotes the projection onto $\mathcal{M}_0$. This projection enhances data consistency by ensuring that the estimated $\hat{\boldsymbol{x}}_0(\boldsymbol{y})$ satisfies the measurement constraints. For this purpose, we adopted Krylov subspace methods by employing a $k$-step Conjugate Gradient (CG) algorithm to efficiently approximate this projection, starting from $\hat{\boldsymbol{x}}_0(\boldsymbol{x}_t, \boldsymbol{y})$:

$$\hat{\boldsymbol{x}}_0(\boldsymbol{y}) \leftarrow \mathrm{CG}(\boldsymbol{A}^T \boldsymbol{A}, \boldsymbol{A}^T \boldsymbol{y}, \hat{\boldsymbol{x}}_0(\boldsymbol{x}_t, \boldsymbol{y}), k) \tag{20}$$

**Forward Sampling** In the previous step, we considered the data enhanced posterior mean $\hat{\boldsymbol{x}}_0(\boldsymbol{y})$ estimation, we now propose to sample to the time $t$ using the forward process of DDPM as shown in (4):

$$\boldsymbol{x}_t(\boldsymbol{y}) := \sqrt{\bar{\alpha}_t}\hat{\boldsymbol{x}}_0(\boldsymbol{y}) + \sqrt{1 - \bar{\alpha}_t}\boldsymbol{z} \tag{21}$$

where $\boldsymbol{z} \sim \mathcal{N}(\boldsymbol{0}, \boldsymbol{I})$. This step effectively samples $\boldsymbol{x}_t(\boldsymbol{y})$ from $p_t(\boldsymbol{x}_t \mid \hat{\boldsymbol{x}}_0(\boldsymbol{y}))$, aligning with the forward process of DDPM and ensuring that the sample corresponds to the appropriate noise level at timestep $t$.

**Reverse Sampling with DDIM** Finally, we perform one step of DDIM reverse sampling to obtain $\boldsymbol{x}_{t-1}$ from $\boldsymbol{x}_t(\boldsymbol{y})$:

$$\boldsymbol{x}_{t-1} := \sqrt{\bar{\alpha}_{t-1}}\hat{\boldsymbol{x}}_0(\boldsymbol{x}_t(\boldsymbol{y})) + \sqrt{1 - \bar{\alpha}_{t-1}} \cdot \boldsymbol{\epsilon}_\theta\left(\boldsymbol{x}_t(\boldsymbol{y}), t\right) \tag{22}$$

where

$$\hat{\boldsymbol{x}}_0(\boldsymbol{x}_t(\boldsymbol{y})) := \frac{\boldsymbol{x}_t(\boldsymbol{y}) - \sqrt{1 - \bar{\alpha}_t}\boldsymbol{\epsilon}_\theta\left(\boldsymbol{x}_t(\boldsymbol{y}), t\right)}{\sqrt{\bar{\alpha}_t}} \tag{23}$$

which ensures that all components in the reverse sampling process remain consistent with the measurement $\boldsymbol{y}$.

**Accelerated Sampling with Improved Initialization** We observe that the estimate $\hat{x}_0$ may deviate significantly from the ground truth when $t$ is large (i.e., at early timesteps in reverse process). Data consistency is more beneficial during the later stages of the sampling process when $t$ approaches zero. Therefore, we propose to start the reverse diffusion from a much smaller timestep $N' < N$ with an appropriate initialization, which significantly reducing the number of reverse diffusion steps in practice.

Specifically, we initialize $x_{N'}$ by sampling from the forward process of DDPM conditioned on an initial estimate, such as the FBP result $x_{\text{FBP}}$:

$$x_{N'} \sim \mathcal{N}(\sqrt{\bar{\alpha}_{N'}} x_{\text{FBP}}, \sqrt{1 - \bar{\alpha}_{N'}} I) \qquad (24)$$

Intuitively, this approach provides a better starting point that incorporates measurement information, which allows to reduce the required number of diffusion sampling steps without sacrificing reconstruction performance.

Overall, our EffiDPSRecon algorithm consists of two main components: (1) a forward diffusion sampling up to timestep $N'$ using the FBP result $x_{N'}$ as a better initialization, and (2) a reverse conditional diffusion down to $t = 0$ with the data enhancement techniques on posterior mean estimation mentioned earlier. The proposed EffiDPSRecon algorithm is summarized in Algorithm 1.

---

**Algorithm 1** EffiDPSRecon

---

**Require:** Number of steps $N', y$
1: $x_{N'} \sim \mathcal{N}(\sqrt{\bar{\alpha}_{N'}} x_{\text{FBP}}, \sqrt{1 - \bar{\alpha}_{N'}} I)$
2: **for** $t = N'$ **to** $0$ **do**
3:     $\hat{\varepsilon}_t = \varepsilon_\theta(x_t, t)$
4:     $\hat{x}_0 \leftarrow \frac{1}{\sqrt{\bar{\alpha}_t}} \left( x_t - \sqrt{1 - \bar{\alpha}_t} \hat{\varepsilon}_t \right)$     ▷ Tweedie's formula
5:     $\hat{x}_0(x_t, y) \leftarrow \hat{x}_0 - \rho_t \nabla_{x_t} \|y - A\hat{x}_0\|_2^2$     ▷ Posterior mean
6:     $\hat{x}_0(y) \leftarrow \text{CG}(A^T A, A^T y, \hat{x}_0(x_t, y), k)$
7:     $z \sim \mathcal{N}(0, I)$
8:     $x_t(y) \leftarrow \sqrt{\bar{\alpha}_t} \hat{x}_0(y) + \sqrt{1 - \bar{\alpha}_t} z$     ▷ Forward sampling
9:     $\hat{x}_0(x_t(y)) \leftarrow \frac{x_t(y) - \sqrt{1 - \bar{\alpha}_t} \epsilon_\theta(x_t(y), t)}{\sqrt{\bar{\alpha}_t}}$
10:     $x_{t-1} \leftarrow \sqrt{\bar{\alpha}_{t-1}} \hat{x}_0(x_t(y)) + \sqrt{1 - \bar{\alpha}_{t-1}} \cdot \epsilon_\theta(x_t(y), t)$     ▷ DDIM sampling
11: **end for**
12: **return** $x_0$

---

### 3.2 CONNECTION TO DPS AND MCG

By choosing $\rho_t = \frac{\zeta}{\sqrt{\bar{\alpha}_{t-1}}}$ and setting $\sigma_{\eta_t} = 0$ in Equation (13), the reverse sampling iteration of DPS can be rewritten as:

$$x_{t-1} = \sqrt{\bar{\alpha}_{t-1}} \hat{x}_0(x_t, y) + \sqrt{1 - \bar{\alpha}_{t-1}} \varepsilon_\theta(x_t, t). \qquad (25)$$

This equation shows that DPS performs reverse sampling by adding noise from the score function approximation to the measurement-consistent posterior mean $\hat{x}_0(x_t, y)$. The MCG method enhances this approach by applying a projection onto the measurement subspace through a one-step gradient update after the DPS update, aiming to boost data consistency.

The workflow of one-time iteration of our EffiDPSRecon method compared with DPS and MCG is shown in Figure 2. The concept of our method relies on a data enhancement on a "predicted" posterior mean and use the reverse DDIM as a "corrector" after re-sampling to appropriate noise levels through the forward process. Intuitively, the estimation of our algorithm aligns better with both forward and backward process as well as the inverse problems. The process is further improved by initializing from a better estimate (e.g., the FBP result) instead of starting from a zero-mean Gaussian, providing a starting point closer to the true data manifold, which potentially beneficial for solving inverse problems.

**Computational Cost Analysis** In diffusion-based CT reconstruction, computational costs mainly arise from two factors: (1) the number of function evaluations (NFE) of the neural network $\epsilon_\theta$, and

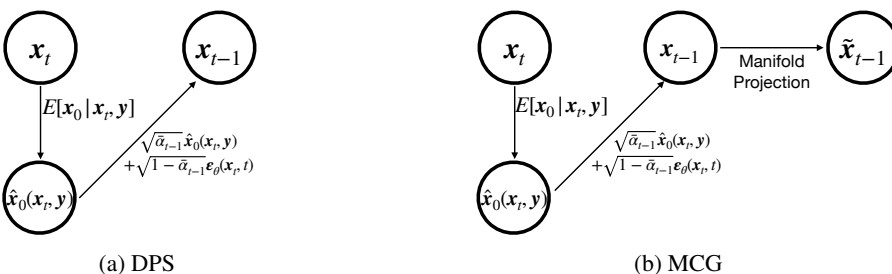

(a) DPS                                        (b) MCG

Figure 2: Diagrams of one iteration in DPS (a), MCG (b), compared to EffiDPSRecon Fig. 1.

(2) the number of Radon transform operations involving $\boldsymbol{A}$ or its adjoint $\boldsymbol{A}^T$. For each iteration, DPS and MCG require one NFE each, while our method requires two occurred in the first posterior mean step and the last DDIM step. Regarding $\boldsymbol{A}$ and $\boldsymbol{A}^T$ operations, DPS requires two per iteration, MCG requires four, and our method requires $2k + 2$, where $k$ is the number of CG iterations in the projection step, which is typically set from 2 to 5.

Although our method incurs a higher per-iteration cost due to the CG computations, it achieves effective posterior sampling in significantly fewer iterations than DPS and MCG (1000 *vs* 50) with data enhancement strategy. This reduction in total iterations offsets the increased per-iteration cost, resulting in overall computational efficiency.

## 4   EXPERIMENTS

### 4.1   DATASETS

The simulated data from human abdomen images provided by Mayo Clinic for the AAPM Low Dose CT Grand Challenge (Moen et al., 2021) are used for evaluation. The dataset contains 2588 NDCT images of thickness 3mm from ten patients resized to $256 \times 256$ resolution. For training, 1923 images from eight patients were used, while the imaging performance was tested on 50 images randomly selected from the remaining two patients. The simulated geometry for projection data includes a flat-panel detector with the source-to-center distance of 535 mm, and the source-to-detector distance of 1024 mm, pixel size of 0.5 mm and 768 detector bins for each projection. The LDCT projections are simulated with both Poisson noise and electronic noise on the corresponding normal dose projection data as follows:

$$\bar{\boldsymbol{y}}_i \sim \text{Poisson}\{I_i \exp(-[\boldsymbol{A}\boldsymbol{x}]_i)\} + \text{Normal}(0, \sigma_e^2), \tag{26}$$

where $\boldsymbol{x}$ denotes the attenuation map with $x_j$ being the linear attenuation coefficient in the $j-$th pixel for $j = 1, \ldots, n$ and $n$ is the total number of pixels. The matrix $\boldsymbol{A}$ is the $m \times n$ system matrix with entries $a_{ij}$, and $[\boldsymbol{A}\boldsymbol{x}]_i = \sum_{j=1}^{n} a_{ij}x_j$ denotes the line integral of the attenuation map $\boldsymbol{x}$ along the $i-$th X-ray with $i = 1, \ldots, m$. $I_i$ is the intensity of incident X-ray incorporating X-ray source illumination and the detector efficiency. $\sigma_e^2$ denotes the electronic noise variance. To reconstruct the attenuation map $\boldsymbol{x}$, we take the logarithm transform on the noisy measurements $\bar{\boldsymbol{y}}$ to generate the noisy sinogram $\boldsymbol{y}$.

We conducted an evaluation of our EffiDPSRecon algorithm under two dose reduction scenarios: (1) Fully-sampled LDCT reconstruction: This scenario involved varying radiation dose levels, specifically $I_i = 10^4, 5 \times 10^4, 10^5$, as outlined in (26); (2) Sparse-view CT reconstruction: In this case, the number of projection views was varied, with configurations set at 32, 64, and 96 views respectively.

In the execution of DDPM framework in our EffiDPSRecon algorithm, we employed a linear sequence for the variance schedule, with the starting and ending values of $\beta_1 = 10^{-4}$ and $\beta_T = 0.02$ respectively. The training is performed using PyTorch interface on a NVIDIA A100 SXM4 80GB GPU. An Adam optimizer is used with the momentum parameter $\beta = 0.99$, mini-batch size set to be 8 and, the learning rate set to be $10^{-4}$. For reconstruction, we choose $N' = 50$ for all the tasks.

## 4.2 METHODS FOR COMPARISON

The performance of the proposed methods is evaluated in comparison to FBP; ADMM-TV; FBPConvNet (Jin et al., 2017) (a DL-based image postprocessing method) and two diffusion based models MCG (Chung et al., 2022), DPS (Chung et al., 2023) (Conditional diffusion model). For the task of LDCT, the regularization parameter $\lambda$ is configured as $5 \times 10^{-4}$ for $I_i = 10^5$ and $I_i = 5 \times 10^4$, and $10^{-3}$ for $I_i = 10^4$. For sparse-view reconstruction tasks, it is set to $10^{-4}$ for 64-view and 96-view, and $5 \times 10^{-3}$ for 32-view. FBPConvNet employs a residual U-Net (Ronneberger et al., 2015) architecture to denoise images reconstructed by the FBP method. The network is trained using the Adam optimizer with a momentum parameter $\beta = 0.99$, a mini-batch size of 4, a learning rate of $10^{-4}$, and for 200 epochs. For MCG and DPS, we use the same hyperparameter and pre-trained checkpoints for diffusion model as our method. The step-size parameter in MCG and DPS is chosen to be $0.1/\nabla_{\boldsymbol{x}_t} \|\boldsymbol{y} - \boldsymbol{A}(\hat{\boldsymbol{x}}_0(\boldsymbol{x}_t))\|_2^2$. The number of diffusion steps is set to be 1000 in order to obtain a high quality sampling results, as also adopted in the original paper of MCG (Chung et al., 2022) and DPS (Chung et al., 2023).

## 4.3 RESULTS

Quantitative assessments with two metrics Peak Signal to Noise Ratio (PSNR) and Structural Similarity (SSIM) of various reconstruction methodologies for LDCT and sparse-view CT datasets are systematically detailed in Table 1. These tables present the mean values of PSNR and SSIM for images reconstructed across varying dose levels and view counts, demonstrating that our method consistently outperforms others across all conditions. Corresponding visualizations of the predicted reconstructed images, generated using these distinct approaches for 32-views and $I_i = 10^4$ LDCT reconstruction are illustrated in Figure 4. Visualizations for other cases are provided in the Appendix (Section A). The display window is set to be [-160,240] HU for all windows with $\mu_{\text{air}} = -1000 HU$. These figures also incorporate difference maps which highlight discrepancies between the reconstructed images and the ground truth data. To provide a closer examination, zoomed-in versions of the images are presented in Figure 5 corresponding to the green zoomed-in box in Figure 4 respectively.

It can be observed that FBP and ADMM-TV methods significantly distort images, severely degrading fine structures and informative features. In contrast, the supervised FBPConvNet method substantially improves the recovery of prominent structures and edges but tends to omit fine details and present blurry boundaries due to missing data in sparse-view projections and noisy data in LDCT projections. Diffusion-based methods like DPS, MCG, and our proposed EffiDPSRecon excel in generation capabilities; however, DPS and MCG struggle to accurately reconstruct structures and fail to capture fine details, as shown in Figure 5. Our EffiDPSRecon method overcomes these challenges, enhancing image fidelity with increased sharpness and detail resolution. This demonstrates the robustness of our algorithm in handling variations in LDCT and sparse-view CT imaging. Visual and quantitative metrics clearly show that our method surpasses other competing CT reconstruction methods in accuracy and image fidelity.

Table 2 displays the computation time required for CT reconstruction using different diffusion methods. Notably, the DPS and MCG methods take at least 5 minutes per slice. However, our algorithm only requires almost 10% computation time of the other two diffusion-based algorithms, while achieving a much better reconstruction quality as shown in Table 1.

| Task | Sparse view CT | | | Low dose CT | | |
|---|---|---|---|---|---|---|
| Methods | 32-views | 64-views | 96-views | $I_i = 10^4$ | $I_i = 5 \times 10^4$ | $I_i = 10^5$ |
| | PSNR/SSIM | PSNR/SSIM | PSNR/SSIM | PSNR/SSIM | PSNR/SSIM | PSNR/SSIM |
| FBP | 22.05/0.3307 | 26.93/0.5186 | 29.58/0.6501 | 28.48/0.5901 | 34.56/0.8495 | 36.52/0.9115 |
| ADMM-TV | 27.09/0.7997 | 31.09/0.8420 | 32.10/0.8495 | 30.69/0.8503 | 35.07/0.9253 | 36.80/0.9436 |
| FBPConvnet | 31.94/0.8301 | 34.82/0.8611 | 35.75/0.9051 | 37.55/0.9108 | 39.33/0.9236 | 40.26/0.9545 |
| DPS (1000) | 37.88/0.9222 | 41.67/0.9618 | 42.33/0.9648 | 33.93/0.8155 | 37.72/0.9050 | 38.53/0.9190 |
| MCG (1000) | 35.80/0.8777 | 40.56/0.9466 | 42.28/0.9620 | 34.57/0.8085 | 38.51/0.9150 | 40.51/0.9451 |
| EffiDPSRecon (50) | **39.88/0.9635** | **45.45/0.9863** | **46.19/0.9883** | **40.39/0.9668** | **43.31/0.9807** | **44.24/0.9838** |
| | 2.00↑/0.0413↑ | 3.78↑/0.0245↑ | 3.86↑/0.0235↑ | 2.84↑/0.0560↑ | 4.80↑/0.0554↑ | 3.73↑/0.0293↑ |

Table 1: Quantitative evaluation of Sparse-view CT and LDCT reconstruction. **Bold**: best.

| Methods | Computation cost / s | | | | | |
|---|---|---|---|---|---|---|
| | 32-views | 64-views | 96-views | $I_i = 10^4$ | $I_i = 5 \times 10^4$ | $I_i = 10^5$ |
| DPS (1000) | 310.18 | 444.95 | 586.25 | 623.28 | 622.19 | 625.14 |
| MCG(1000) | 474.68 | 754.08 | 1042.13 | 1108.32 | 1110.28 | 1111.41 |
| EffiDPSRecon (50) | **45.26** | **71.48** | **85.21** | **58.09** | **59.81** | **58.56** |

Table 2: Computation cost for different diffusion methods. **Bold**: best.

Additionally, we implemented a consistent acceleration strategy across our method and the diffusion-based DPS and MCG methods by setting an identical number of reverse sampling steps, $N'$, as defined in Equation 24. This strategy was evaluated in scenarios such as sparse-view reconstruction with 32 angles and LDCT imaging, where $I_i = 10^4$, illustrated in Figures 3. Our objective was to assess whether the acceleration strategy is equally effective for MCG and DPS as it is for our method. The results indicate that while our method retains its performance robustly with reductions in $N'$ to 10, 20, 50, and 100, the performance of both DPS and MCG significantly deteriorates.

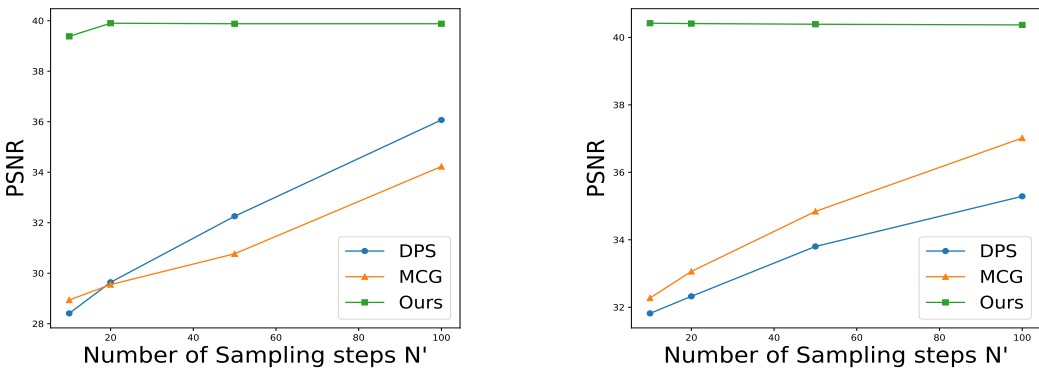

Figure 3: Impact of Sampling Steps ($N'$) on 32-View (left) and LDCT with $I_i = 10^4$ (right) Reconstruction Using Diffusion-Based Methods with the same FBP initialization.

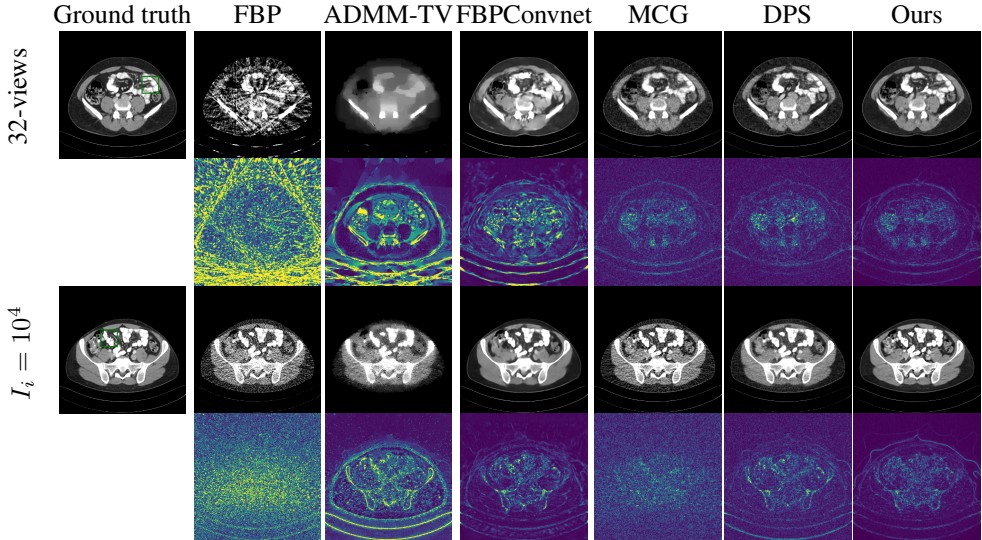

Figure 4: Reconstruction results and their associated absolute difference map for Sparse-view CT and LDCT. The display window is [-160, 240] HU.

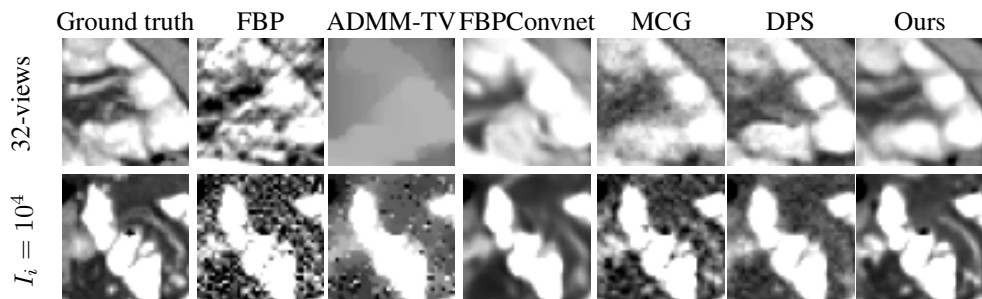

Figure 5: Zoomed-in results of Sparse-view CT and LDCT reconstruction in Figure 4

| Methods | CG | Forward sampling | 32-views | | $I_i = 10^4$ | |
|---|---|---|---|---|---|---|
| | | | PSNR | SSIM | PSNR | SSIM |
| $\mathcal{S}_1$ | ✓ | ✗ | 37.41 | 0.9465 | 37.45 | 0.9541 |
| $\mathcal{S}_2$ | ✗ | ✓ | 33.37 | 0.8645 | 37.55 | 0.9576 |
| Ours | ✓ | ✓ | **39.88** | **0.9635** | **40.39** | **0.9668** |

Table 3: Ablation study for the effect of CG and forward sampling step with 32 views and $I_i = 10^4$ using EffiDPSRecon.

### 4.4 ABLATION STUDY

In the ablation study, we assess the impact of CG and the forward sampling step, specifically in scenarios of sparse-view CT reconstruction with 32 angles and LDCT where $I_i = 10^4$. We establish two baselines, $\mathcal{S}_1$ and $\mathcal{S}_2$, to compare models lacking either CG or the forward sampling step. Table 3 quantitatively contrasts performances, revealing that incorporating both CG and the forward sampling step significantly enhances EffiDPSRecon. Notably, improvements in both PSNR and SSIM metrics highlight the benefits of this integration.

## 5 CONCLUSION

In this paper, we proposed an efficient diffusion posterior sampling scheme for CT image reconstruction (EffiDPSRecon). Experimental results demonstrate that our EffiDPSRecon method can effectively produce high-quality reconstructions from both LDCT and Sparse-view CT. Future work will aim to explore the theoretical property of the proposed method and explore its application in other imaging inverse problems for a broader range of applications.

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

# A    ADDITIONAL EXPERIMENTS

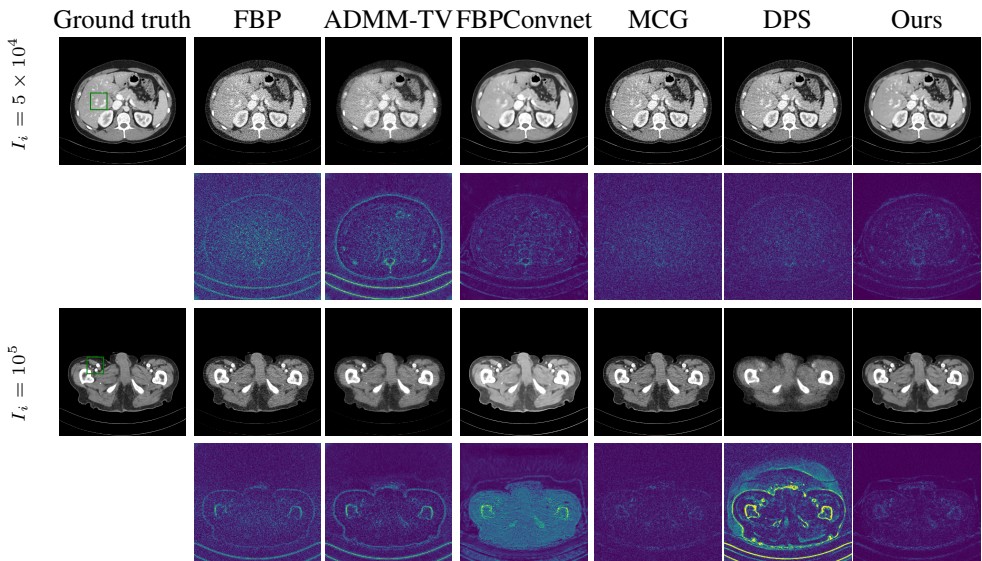

Figure 6: Reconstruction results and their associated absolute difference map for LDCT reconstruction. The display window is [-160, 240] HU.

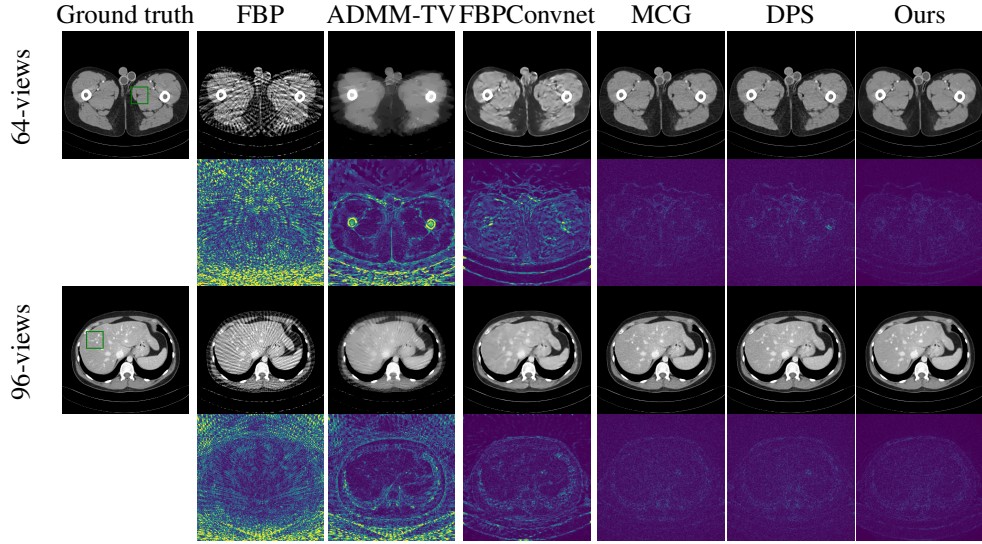

Figure 7: Reconstruction results and their associated absolute difference map for Sparse-view CT reconstruction. The display window is [-160, 240] HU.

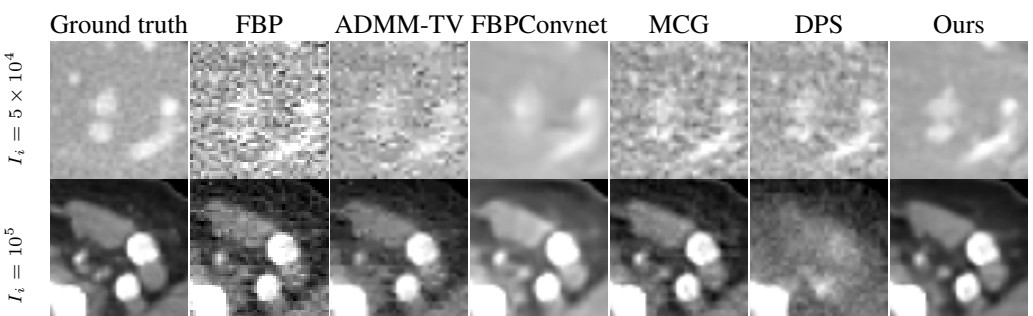

Figure 8: Zoomed-in results of LDCT reconstruction in Figure 6

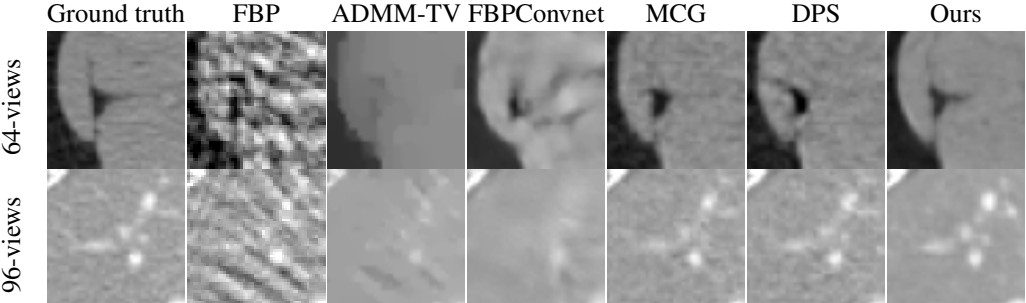

Figure 9: Zoomed-in results of Sparse-view CT reconstruction in Figure 7

