# OpenReview forum: "Efficient Diffusion Posterior Sampling for Dose Reduced CT Reconstruction"
_ICLR.cc/2025/Conference — ICLR 2025 Conference Withdrawn Submission_

### Official Review · Reviewer_WfST · 2024-10-28

**Soundness:** 3
**Presentation:** 2
**Contribution:** 1
**Rating:** 3
**Confidence:** 5

**Summary:**

In this paper, the authors propose an efficient diffusion-based posterior sampling strategy for dose reduced CT reconstruction. Improved data consistency is achieved by incorporating measurement information across multiple stages of the diffusion sampling process, combined with posterior mean estimation through single-step gradient updates and data-enhancement projections using a multi-step conjugate gradient algorithm. Additionally, the authors introduce forward sampling and small time-step initialization strategies to significantly accelerate the sampling speed. Simulation results demonstrate that this method provides higher-quality reconstructions in shorter sampling times.

**Strengths:**

- Regarding the results shown in the paper, the proposed method achieves higher-quality reconstruction with shorter sampling time.

**Weaknesses:**

### Major Weaknesses:
1. The proposed method essentially consists of four main components: (1) _posterior mean estimation_ and (2) _data-enhancement projections_ to ensure data consistency, (3) _forward sampling_ and (4) _small time-step initialization_ to accelerate sampling. However, component (1) is actually the DPS [1] method, (2) is actually the DDS [2] method, and (4) is widely recognized within the community as originally derived from CCDF [3]. Thus, the approach in this paper is essentially a combination of these three works, has not shown any theoretical progress, and lacks the level of novelty expected by the ICLR community. Even more surprising is that references [2] and [3] are not cited at all in this paper.
2. Most directly, the core of this paper is derived from DDS [2]; however, it lacks a comparison with DDS, either through theoretical analysis or experimental evaluation.
3. Stepping back, even within the exploration of combining the two diffusion-based inverse problem solver paradigms, DPS and DDS, more refined and complete work has emerged [4], which further diminishes the contribution of this paper.

### Minor Weaknesses:
1. The sentence in the abstract, '..._by an average of 3.5 **db** but also accelerates_...,' incorrectly uses all lowercase for the unit of decibels, which should be written as 'dB.'
2. Punctuation is missing after some formulas, e.g., Eq. 21, etc.
3. The notation used in Algorithm 1 lacks explanation and should be clearly defined to enhance the reader's understanding.

### Reference
- [1] Chung, Hyungjin, et al. "Diffusion Posterior Sampling for General Noisy Inverse Problems." The Eleventh International Conference on Learning Representations.
- [2] Chung, Hyungjin, Suhyeon Lee, and Jong Chul Ye. "Decomposed Diffusion Sampler for Accelerating Large-Scale Inverse Problems." The Twelfth International Conference on Learning Representations.
- [3] Chung, Hyungjin, Byeongsu Sim, and Jong Chul Ye. "Come-closer-diffuse-faster: Accelerating conditional diffusion models for inverse problems through stochastic contraction." Proceedings of the IEEE/CVF Conference on Computer Vision and Pattern Recognition. 2022.
- [4] Askari, Hossein, Fred Roosta, and Hongfu Sun. "Bi-level Guided Diffusion Models for Zero-Shot Medical Imaging Inverse Problems." arXiv preprint arXiv:2404.03706 (2024).

**Questions:**

1. Chung et al. state in the DDS article that "..._if the tangent space at each denoised sample is representable by a Krylov subspace, there’s no need to compute the MCG. Rather, the standard CG method suffices to guarantee that the updated samples stay within the tangent space._" Could the authors provide insights into the benefits of using both MCG and CG? For instance, they could offer a theoretical explanation or include ablation experiments to illustrate these advantages.
2. Table 2 shows that for the LDCT task, the computation time of the proposed method is significantly shorter than that of the comparison method and even much shorter than that of SVCT with 96 views. This seems counterintuitive. In the LDCT task, you should use full-view instead of SVCT, right? With radon and back-projection operations performed in CG, the process should theoretically be slower than SVCT.

---

### Official Review · Reviewer_tkhq · 2024-11-02

**Soundness:** 3
**Presentation:** 3
**Contribution:** 2
**Rating:** 6
**Confidence:** 4

**Summary:**

This paper proposes a new posterior sampling method for a pre-trained diffusion model to be applied to imaging inverse problems. The approach uses a combination of posterior mean estimation and data consistency projection at each time step to produce samples that adhere to the measurements. It then involves a forward resampling step to ensure that the projected sample is compatible with the noisy data space, preparing it for the reverse denoising process. To reduce the required sampling steps, the method is initialized with Filtered Backprojection images instead of noise.

**Strengths:**

- The method seems reasonable and well-designed.
- The paper nicely provides a section (3.2) summarizing its advantages compared to MCG and DPS.
- The performance is impressive compared to other diffusion-based methods.

**Weaknesses:**

1. The idea of initializing with a FBP reconstructed image instead of random noise to accelerate sampling is not new.
2. Using conjugate gradient descent increases the per-iteration computational cost. While initializing with an FBP image reduces the required iterations, similar modifications could also be applied to DPS and MCG to reduce the sampling steps.
3. The training and evaluation datasets seem quite small: only 8 subjects for training, and only 50 images from the 2 remaining patients for testing. There could be significant overlap among these 50 slices.

**Questions:**

See above.

---

### Official Review · Reviewer_bFmG · 2024-11-03

**Soundness:** 2
**Presentation:** 3
**Contribution:** 2
**Rating:** 3
**Confidence:** 5

**Summary:**

This paper proposes a diffusion-based method for dose-reduced CT reconstruction. This method utilizes the projections to guide the denoising direction and utilizes DDIM to accelerate the denoising process.

**Strengths:**

+This method leverages the projections to guide the denoising (image generation) process. Compared to direct image generation using diffusion models, this approach yields more reliable results.
+Compared to other diffusion-based CT reconstruction algorithms (MCG and DPS), this method utilize DDIM and achieves faster reconstruction results with fewer denoising iterations.

**Weaknesses:**

- It is based on MCG and utilizes DDIM for acceleration.
- The experiments were conducted only on a single dataset (AAPM), and generalizability cannot be guaranteed. I am curious about the performance of the model trained on AAPM when tested on other datasets.
- The experiments were based on dose simulations derived from AAPM data, and there were no tests conducted on real low-dose images, which may compromise its real-world efficacy.
- Compared to other methods, the diffusion-based CT reconstruction approach has significant advantages in extremely sparse-view [1] (2 and 8 views) or extremely low doses [2] (5% dose), achieving better performance. However, the experiments in this paper did not include settings for such scenarios.

[1] Liu J, Anirudh R, Thiagarajan J J, et al. DOLCE: A model-based probabilistic diffusion framework for limited-angle ct reconstruction. In Proceedings of the IEEE/CVF International Conference on Computer Vision. 2023.

[2] Gao Q, Li Z, Zhang J, et al. CoreDiff: Contextual error-modulated generalized diffusion model for low-dose CT denoising and generalization. IEEE Transactions on Medical Imaging, 2023.

**Questions:**

1. what other innovations does this method have compared to MCG besides using DDIM for acceleration and CG to approximate the projections?
2. More comprehensive and thorough experiments are essential.

---

### Official Review · Reviewer_9Fu7 · 2024-11-03

**Soundness:** 2
**Presentation:** 3
**Contribution:** 2
**Rating:** 3
**Confidence:** 3

**Summary:**

Submission 13495 presents a method for accelerating CT reconstruction when using diffusion model-based reconstruction with a form of data consistency and better initialization. Current methods require substantial amounts of sampling iterations and sometimes hallucinate details. The submission enforces consistency of the reconstructions with the sensor domain measurements, thus improving image fidelity. Further, it finds that using a standard CT reconstruction (filtered back projection) as initialization can significantly speed up the reconstruction.

**Strengths:**

- The paper is well organized and a quick read.
- The paper does a good job of summarizing previous DDPM work and relating it to its own contributions in a somewhat self-contained manner.
- The ideas presented herein are quite intuitive and seemingly useful.

**Weaknesses:**

# 1. Methodological contributions unclear

The paper has two main technical contributions: data consistency during reconstruction and the use of FBP initialization to accelerate the reconstruction. Both have significant uncited overlap with previous work as detailed below.

## 1.1. Data-consistent diffusion-based reconstruction already studied

The paper’s main contribution is the use of data consistency during reconstruction. However, [ReSample](https://arxiv.org/pdf/2307.08123) (ICLR 2024 spotlight) already introduces the notion of hard data consistency and uses conjugate gradient methods that are quite similar to those presented in this submission. In fairness, ReSample enforces data consistency in latent space, whereas it appears to be enforced in pixel space here.

Could the authors please detail why ReSample and similar works were not cited and/or benchmarked against in this submission? Further, it would be great to have an enumerated list of differences alongside a comparison.

## 1.2. Claimed FBP initialization contribution is widely used

The submission initializes the diffusion model using the FBP reconstruction as a starting point which is the key ingredient in accelerating the reconstruction. However, FBP-based initialization is universally used as a starting point for all iterative reconstruction and DL-based methods (including on commercially deployed scanners, see [page 6](https://www.cockcroft.ac.uk/wp-content/uploads/2016/04/CT-Iterative_Reconstruction_Techniques.pdf) ), so it is unclear what the specific contribution is here. Could the authors please elaborate on this?

# 2. Experimental issues

## 2.1. No held-out test data or use of validation sets / Tiny test sets

The paper states that 10 subjects were chosen from a simulated dataset (more on that below) and 8 were used for training and 2 were used for testing.
- As it is not mentioned, this implies that a validation set was not used and that the test set was used as is for model development thus leading to overly optimistic results for the proposed method. If a validation set was used, please clarify.
- _Two_ testing subjects and a single dataset constitute far too small of a sample size to support any claims. I understand that each subject had multiple simulated X-rays, however, this does not guard against the extreme inter-slice correlation within a single subject. As detailed below, there are several (non-simulated) datasets that could be used for benchmarking. For example, ReSample uses 40 real subjects with real projection data.

Please elaborate on why such a limited evaluation set was used.

## 2.2. Only simulated data

While this is endemic across the field, the submission uses _only_ simulated synthetic X-ray projection data in its experiments, simulating it using the same exact forward model as it does in its model. As per the “inverse crime” phenomenon, this can create highly optimistic results and exaggerate differences between methods.

Within CT, there is a small set of datasets that provide both CT and raw _measured_ projection data. For example, please see:
- https://www.cancerimagingarchive.net/collection/ldct-and-projection-data/ (they provide scripts to rebin to fan-beam if necessary. This dataset was also used in the hard data consistency paper referenced above)
- https://www.nature.com/articles/s41597-019-0235-y
- https://www.nature.com/articles/s41597-023-02484-6

Could the authors please detail why the experiments only use simulated projections?

## 2.3. No ablation for FBP initialization

The FBP initialization is claimed as a contribution, however, it is not ablated. As it is likely that the proposed method would slow down significantly without it, please include an ablation studying its effect.

## 2.4. Misc. baseline issues
- The ADMM-TV baseline is producing results that are clearly over-regularized. Was its TV weight not swept?
- In section 4.2, the paper’s methods have their hyperparameters carefully tuned, whereas MCG and DPS (the two main baselines) appear to not have their hyperparams tuned and are matched to the paper’s settings. Could you elaborate on why this would be optimal for them?

# Minor
- Please increase the length of the x-axes in Fig3 from 1–100 to 1–1000 s.t. readers can assess when DPS and MCG saturate.
- Table 2 is confusing. Did the authors mean “Computation cost (s)” instead of “Computation cost/s”? Otherwise, I do not understand what is being reported.
- The apparent use of LLMs for editing writing in this paper has made it quite hard to read. For example, the abstract is hard to parse and understand what is being technically conveyed. Another editing pass with humans would improve it significantly.

**Questions:**

- Could the authors please detail why ReSample and similar works were not cited and/or benchmarked against in this submission? Further, it would be great to have an enumerated list of differences alongside a comparison.
- Could the authors please elaborate on how FBP initialization is a new contribution if it is widely used?
- Could the authors please elaborate on why such a limited evaluation set was used in this work? Further, why was a validation set not used?
- Could the authors please detail why the experiments only use simulated projections?
- Can the rebuttal provide a speed and fidelity based ablation for FBP initialization?

---

### Note · Authors · 2024-11-25

**Comment:**

After a careful review of the feedback from the reviewers, we decide to withdraw our paper to further refine the methodology and address the concerns raised during the review process.

Upon reviewing the feedback from Reviewers 9Fu7 and WfST, we recognize that an oversight occurred in our literature review process and we failed to conduct a thorough study of related work, which resulted in the absence of important citations and comparison in our submission. This lack of inadvertently led to overlaps in novelty and practical contributions with previously established methodologies [1][2][3] that were not properly acknowledged in our manuscript. We sincerely apologize for this oversight. Moving forward, we are committed to significantly improving this aspect of our work by enhancing our literature review and expanding the comparisons with existing methods. However, we would still like to emphasize the difference to the other work and the contributions of our paper:

(1)Compared to ReSample [1], our method incorporates an extra forward sampling step at $t+1$ where $\hat x_0(x_t,y)$ denoised via Tweedie's formula undergo data consistency optimization, are then re-noised to $x_{t+1}(y)$ for an extra unconditional reverse sampling to get $x_t$. This differs from the ReSample [1] approach where the denoised output $\hat z_0(y)$ optimized after Tweedie's formula is directly re-noised to $z_t$ using a new sampling strategy with reduced variance (Proposition 2) in the latent space. Due to the absence of pre-trained model parameters and training details for the CT medical imaging in the official source code, as well as the fact that the whole Resample algorithm is implemented in the latent space, we are unable to provide an immediate comparative results. However, we plan to include comparative experiments in the future to refine our work.

(2)Using FBP for initialization to reduce sampling steps is a natural idea for acceleration despite the existence of CCDF work [3]; On the other hand, as shown in Figure 10 in our paper, our method achieves more stable and better reconstruction results compared to other methods even when employing the same initialization with the same number of sampling steps.

(3)The conjugate gradient (CG) algorithm in our paper has indeed been used in DDS [2]. However, the forward sampling step also plays an important role in enhancing data consistency in our proposed algorithm. In our paper, we employed the Operator Discretization Library (ODL) framework for Radon and FBP transformations which is different from the implementation of previous work. Thus, for a fair comparison with DDS, we used the same pretrained checkpoints and test images of DDS in the official GitHub website https://github.com/HJ-harry/DDS for 2D sparse-view reconstruction and adopted 1000-step sampling. The comparison results were as follows:

8-view Reconstruction: PSNR: DDS = 29.87, Our method = 30.03;

SSIM: DDS = 0.8314, Our method = 0.8664.

4-view Reconstruction: PSNR: DDS = 29.20, Our method = 29.47;

SSIM: DDS = 0.8185, Our method= 0.8544.

 These results demonstrate our method's superiority in both PSNR and SSIM metrics, underscoring the role of other components in our method.

For the suggestions regarding datasets and experiments provided by Reviewers 9Fu7 and bFmG, we plan to conduct a more comprehensive set of experiments, including the use of real-world data, to validate our approaches. It is crucial to acknowledge that in the field of CT reconstruction, even when studies utilize real CT data, the projections are typically simulated. This commonly occurs because real projection operators, which are essential for accurately solving inverse problems, are seldom available. This discrepancy between the availability of real projection data and the lack of corresponding projection operators poses challenges in accurately modeling and solving these inverse problems.  In our revised work, we will address these challenges and ensure thorough comparisons and citations of related works to clearly position our contributions within the existing research landscape.

We appreciate the opportunity to participate in the ICLR process, and we are grateful for the detailed and constructive critiques provided by the reviewers. These comments will guide our ongoing research, and we look forward to presenting an improved version of our work in the future.

Reference

[1] Song, B., Kwon, S. M., Zhang, Z., Hu, X., Qu, Q., & Shen, L. Solving Inverse Problems with Latent Diffusion Models via Hard Data Consistency. In The Twelfth International Conference on Learning Representations, 2024.

[2] Chung, H., Lee, S., & Ye, J. C. Decomposed Diffusion Sampler for Accelerating Large-Scale Inverse Problems. In The Twelfth International Conference on Learning Representations, 2024.

[3] Chung, H., Sim, B., & Ye, J. C.  "Come-closer-diffuse-faster: Accelerating conditional diffusion models for inverse problems through stochastic contraction." Proceedings of the IEEE/CVF Conference on Computer Vision and Pattern Recognition. 2022.

**Withdrawal Confirmation:**

I have read and agree with the venue's withdrawal policy on behalf of myself and my co-authors.